# MRI Spinal Cord Reconstruction Provides Insights into Mapping and Migration Following Percutaneous Epidural Stimulation Implantation in Spinal Cord Injury

**DOI:** 10.3390/jcm13226826

**Published:** 2024-11-13

**Authors:** Siddharth Venigalla, Muhammad Uzair Rehman, Jakob N. Deitrich, Robert Trainer, Ashraf S. Gorgey

**Affiliations:** 1Spinal Cord Injury and Disorders, Richmond VA Medical Center, Richmond, VA 23249, USA; 2Department of Biomedical Engineering, Virginia Commonwealth University, Richmond, VA 23284, USA; 3Physical Medicine and Rehabilitation, Richmond VA Medical Center, Richmond, VA 23249, USA; 4Physical Medicine and Rehabilitation, School of Medicine, Virginia Commonwealth University, Richmond, VA 23298, USA

**Keywords:** spinal cord epidural stimulation (SCES), temporary mapping, permanent mapping, MRI, spinal cord reconstruction, spinal cord injury

## Abstract

**Background**: Spinal cord epidural stimulation (SCES) has the potential to restore motor functions following spinal cord injury (SCI). Spinal cord mapping is a cornerstone step towards successfully configuring SCES to improve motor function, aiming to restore standing and stepping abilities in individuals with SCI. While some centers have advocated for the use of intraoperative mapping to anatomically target the spinal cord locomotor centers, this is a resource-intensive endeavor and may not be a feasible approach in all centers. **Methods**: Two participants underwent percutaneous SCES implantation as part of a clinical trial. Each participant underwent a temporary (1-week, two-lead) trial followed by a permanent, two-lead implantation. SCES configurations were matched between temporary and permanent mappings, and motor evoked potential in response to 2 Hz, for a duration of 250–1000 µs and with an amplitude of 1–14 mA, was measured using electromyography. T2 axial MRI images captured prior to implantation were used to retrospectively reconstruct the lumbosacral segments of the spinal cord. The effects of lead migration on mapping were further determined in one of the participants. **Results**: In both participants, there were recognized discrepancies in the recruitment curves of the motor evoked potentials across different muscle groups between temporary and permanent SCES mappings. These may be explained by retrospective MRI reconstruction of the spinal cord, which indicated that the percutaneous leads did not specifically target the entire L1-S2 segments in both participants. Minor lead migration appeared to have a minimal impact on spinal cord mapping outcomes in one of the participants but did dampen the motor activity of the hip and knee muscle groups. **Conclusions**: Temporary mapping coupled with MRI reconstruction has the potential to be considered as guidance for permanent implantation considering target activation of the spinal cord locomotor centers. Since lead migration may alter the synergistic coordination between different muscle groups and since lead migration of 1–2 contacts is expected and planned for in clinical practice, it can be better guided with proper spinal cord mapping and a diligent SCES lead trial beforehand.

## 1. Introduction

Spinal cord injury (SCI) is a debilitating neurological state primarily caused by vertebral fracture or dislocation leading to neuronal damage [1]. What ensues is glial scarring and inflammation at the site of injury [2]. SCI is associated with a range of comorbidities, including limited functional mobility, respiratory dysfunction, cardiovascular dysfunction, depression, sexual dysfunction, and bladder and bowel issues [3]. SCI also poses a significant financial burden on healthcare, with one study estimating the lifetime expenditure per individual with SCI to be USD 0.7–2.5 million [4]. Several interventions have been developed to treat SCI, such as the use of omega-3 fatty acids, which have shown potential for their anti-inflammatory and neurogenerative effects but require further validation in multicenter studies [5]. Other interventions include the use of spinal cord stimulation to help treat injury-associated chronic pain, which is promising but also requires further evidence on its effectiveness from future studies [6].

Spinal cord stimulation is encompassed by the science of neuromodulation, which has progressed to cover a variety of medical conditions, including chronic pain, spasticity, pulmonary functions and bladder control [7,8,9,10]. This progression has led researchers to conclude that spinal cord stimulation (SCS), delivered by paddles or leads implanted into the epidural space, is an effective modality for the restoration of motor functions after spinal cord injury (SCI) [11,12,13,14,15]. Seminal studies indicated restoration of overground stepping in persons with motor-complete SCI [11,12,13,14]. Wagner et al. showed that customized closed-loop spinal cord epidural stimulation (SCES) would restore overground walking within a few days post-implantation [12]. Others also noted that SCES is beneficial in the regulation of autonomic nervous system dysfunction either during postural changes or during exercise [16,17,18]. Despite the use of different SCES technologies, research studies agreed that spinal cord mapping is a cornerstone step to achieve functional motor recovery [19,20,21,22,23].

The potential of using percutaneous SCES to promote motor recovery was recently explored in men with SCI [15,20,23,24,25]. Several reports indicated that percutaneous leads may restore trunk control, standing and overground non-functional stepping [15,23,25]. Furthermore, the authors noted that using interleaved configurations is likely to reduce spastic tone and enhance overground non-functional stepping in a person with SCI [23]. Recently, we explored the potential of using the peak slope ratio to optimize the SCES configurations necessary to achieve standing in men with SCI [20]. Compared to simply using motor evoked potentials (muscle action potentials elicited by spinal cord stimulation), the slope ratios of the established recruitment curves (motor evoked potentials plotted versus stimulation amplitude) identified and narrowed down the best stimulation parameters as well as the cathodal–anodal placements in the form of wide- vs. narrow-field configurations [20]. For a myriad of reasons, including ASIA level and overall physical conditioning, only two men out of the four participants were capable of achieving independent standing [20]. As a follow-up, we were interested to further understand why functional standing was not achieved in these participants.

Earlier work demonstrated that successful anatomical placement of a surgically implanted paddle relies on the process of intraoperative mapping [19,22]. Mapping refers to the process of testing various stimulation configurations to achieve targeted activation of specific spinal motor pools [13,19,22]. Intraoperative mapping is a method commonly used to identify the optimal location for the placement of the electrodes corresponding to the spinal cord segments [13,19,22,23]. Previous work highlighted the significance of intraoperative mapping of the spinal cord by referring to specific targeted stimulation of the muscles compared to non-specific stimulation [24]. The authors highlighted step-by-step processes to maximize spinal cord mapping of the posterior lumbosacral nerve roots [24]. The authors determined the motor evoked response (electrical muscle activity due to SCES) of cathodal locations based on the segmental innervation of the rectus femoris (RF; L2-L4) and triceps surae (TS; L5-S2) muscles [24]. Selective target stimulation was based on accurately placing the cathode at L2 and L5 for RF and TS, respectively. Placement of the cathode at L4 led to non-selective stimulation of both muscles. Furthermore, no response in 72 attempts was achieved in both muscles when the cathode was placed corresponding to the L1 segment [24]. A recent review highlighted the significance of target stimulation to achieve motor control in persons with SCI [19]. The authors stressed that magnetic resonance imaging (MRI) should be conducted prior to surgical implantation to perform 3D construction of the cord to determine the location of the conus termination relative to the T12-L2 vertebrae [26]. The rationale was based on a prior study which noted variability in the length and volume of the spinal cord lumbosacral segments as well as the location of conus termination relative to the vertebral bodies [27,28,29]. Non-target placement of the paddle may lead to non-specific stimulation of the spinal cord segments and failure to achieve specific motor activation [24]. The use of MRI spinal cord reconstruction will guide the surgeon in accurate anatomical placements of the paddle in the epidural space relative to the target lumbosacral segments (L1-S2), and this was then confirmed with intraoperative mapping [19,26].

Reliable intraoperative mapping is not available at all centers, and we chose trial mapping in a controlled setting, as is commonly performed when SCS is used for pain management [7,30,31,32]. In common clinical practice, the trial phase ensures that patients successfully experience a reduction in their pain symptoms prior to costly permanent implantation. The implantation proceeds initially with a trial phase (temporary implantation) and is followed by permanent implantation 2–3 weeks later [15,20]. Our center adopted the SCES temporary implantation to ensure the selection of appropriate candidates prior to proceeding with full surgical permanent implantation [10,15,20]. Through the use of a trial, we could offer another chance for consent before the occurrence of permanent surgical implantation and to confirm the ability of threading the leads in the epidural space (for instance, in case of the presence of hardware spinal fusion in this region in persons with SCI). However, it is unclear whether temporary implantation would guide the medical team on how to accurately place the leads during permanent implantation and serve as an alternative for intraoperative mapping. Furthermore, it is unclear whether performing MRI to establish spinal cord reconstruction would provide interpretation of the discrepancy, if any, in motor evoked potentials between temporary and permanent implantations or following migration in persons with SCI. The use of MRI would ensure custom-based implantation as a result of differences in the size and thickness of the lumbosacral segments.

Therefore, the major goal of the current work was to retrospectively determine the potential of using temporary implantation to guide permanent implantation in two persons with SCI after MRI reconstruction of the spinal cord. Additionally, it is unclear whether MRI reconstruction of the lumbosacral segment may explain how migration of the leads may impact motor evoked potential and result in loss of functional outcomes.

## 2. Methods

### 2.1. Human Subjects

Two men with clinically complete traumatic SCI (T6, AIS A, 12 years post-injury [I.D. #: 0883] and T4, AIS A, 24 years post-injury [I.D. #: 0884]) participated in a clinical trial that was approved by our local Medical Center [20]. The timeline of the study protocol for both participants is listed in Figure 1. The participants were provided with written informed consent that explained the purpose of the study, group assignment phases of implantation, SCES stimulation and training throughout the study (Figure 1). The entire study was approved by the institutional review board at the Richmond VA Medical Center (code: 02595, approved on 14 July 2020). All methods were performed as per the relevant guidelines and regulations, and the trial was registered at clinicaltrials.gov, with the registration I.D. # NCT04782947. All data were retrospectively analyzed to address the major goals of the trial.

### 2.2. Timeline of the Study

The participants were enrolled in the study for approximately 12 months, with the purpose of examining the effects of resistance training (RT) and percutaneous SCES (Rest SCI) on the restoration of functional motor control, such as standing and stepping, in persons with SCI (Figure 1). The first 6 months were targeted towards achieving standing, and the remaining 6 months were focused on overground locomotion. The participants participated in three sessions per week, each involving 1 h of exoskeletal-assisted walking (EAW); this was followed by a standing training program, with each session lasting 1 h in duration. The two participants were initially randomized into two groups: pSCES + EAW + RT (0884) or delayed pSCES + EAW + no RT (0883). The delayed pSCES group underwent implantation 6 months after undergoing 6 months of EAW. This group served as a control group to determine the effects of percutaneous SCES on the various parameters of the study. The participants had measurements taken at baseline (prior to training), post-measurement 1 (at the end of the first 24 weeks) and post-measurement 2 (at the end of the second 24 weeks).

### 2.3. Inclusion and Exclusion Criteria

The inclusion and exclusion criteria were previously listed in detail [15]. Briefly, persons with a traumatic, motor-complete SCI with a neurological level of injury C6 and with an age between 18 and 60 years were considered. A certified SCI physician conducted the International Standards for Neurological Classification of SCI (ISNCSCI) to determine the neurological level and severity of the injury. Only participants with an AIS A or B injury (indicating a motor deficit below the level of injury) were included. AIS A and B were only considered to detect the restoration of any subsequent motor control below the level of injury. Finally, the participants had to be 24 months post-SCI to account for possible spontaneous recovery that may have occurred during the initial phase of injury.

### 2.4. Magnetic Resonance Imaging (MRI)

Before enrollment, the participants underwent magnetic resonance imaging (MRI; T2 Turbo Spin Echo with a long band width; SIEMENS 1.5T) with the following scanning sequence: slice thickness: 3 mm; TR: 9350 ms; TE: 102 ms; flip angle: 150, for pre-screening purposes. The initial purpose of the MRI was to confirm the location of the injury and assess the extent of injury, as well as to determine the patency of the lumbosacral region before implantation. Retrospectively, MRI was then utilized as a tool to guide reconstruction of the spinal cord and ensure accurate anatomical placement of the percutaneous leads in the lumbosacral segment.

### 2.5. Implantation of Percutaneous SCES

The trial involved both temporary and permanent implantation (3–4 weeks following temporary implantation) [15,20]. Temporary implantation was performed prior to permanent implantation and lasted 5 days before removal of the leads under fluoroscopy guidance (Figure 2). The SCES system (Intellis Epidural Stimulator; Medtronic, Minneapolis, MN, USA) delivered electrical stimulus to the cord. Before both temporary and permanent proceduresHibiclens^®^ (Norcross, GA, USA) (chlorhexidine) soap skin cleanser and Bactroban^®^ (GlaxoSmithKline, Brentford, UK) (mupirocin) 2% ointment were administered for 7 days before the procedure to reduce the presence of bacteria. Antibiotics (typically, Ancef (2–3 g) or Clindamycin (600–900 mg)) were administered for each procedure. An anesthesia preoperative evaluation was performed, and consent was obtained prior to the implantation procedures. Temporary implantation was conducted to determine possible unforeseen complications that may have led to withdrawal from the trial or qualified the participant as a screen failure.

### 2.6. Temporary Implantation

Implantation was performed in a minor procedure room with the participant in the prone position [15]. During the procedure, a nurse gained IV access and established continuous monitoring of vital signs (standard ASA monitors, including non-invasive blood pressure every 5 min, pulse oximetry, continuous EKG and end-tidal CO2 from a nasal cannula). The epidural space was accessed through 14-gauge epidural needles using the loss-of-resistance technique with X-ray assistance. Leads were then placed in the epidural space, and the configuration necessary to evoke motor potentials was tested, as indicated by visible motor contractions of the paralyzed muscles. A Medtronic representative was present to oversee the entire procedure. The leads were guided to both sides of the midline on live fluoroscopy to confirm posterior epidural placement. Following initial testing via fluoroscopy, the lead position was optimized after confirmation of proper motor stimulation. After the lead placement was finalized, the needles in the epidural space were removed. Electrodes were secured to the skin with tape and glue to decrease the possibility of lead migration. EMG spinal mapping was performed on the same day or the following day for 3 consecutive days.

### 2.7. Permanent Implantation

After temporary implantation confirmed correct activation of the lumbosacral segments and the participant was given a second chance to consent, we then proceeded with permanent implantation. Two 8-electrode arrays of Vectris leads were implanted 3–4 weeks after the temporary implantation [15]. The only difference from the first phase of temporary implantation was that an anesthesiologist performed IV sedation, and standard ASA monitors were again used during the procedure. The epidural space was again accessed through 14-gauge epidural needles using the loss-of-resistance technique with X-ray guidance. Leads were then placed in the epidural space, and the configuration necessary to evoke motor potentials was retested, as indicated by visible motor contractions of the paralyzed muscles.

After a vertical incision was made in the lateral low back between the 12th rib and the iliac crest, a pocket was dug out between the skin and muscle for the battery with a bovie and scissors, and the pulse generator was then placed in the pocket. The leads were secured with non-absorbable 0 monocryl sutures to the interspinal ligament and/or the lumbodorsal fascia. Next, the leads were guided under the skin to the pocket of tissue, where they were then connected to a Medtronic Intellis battery. Impedances were checked after hemostasis was complete and irrigation applied. The wound was sealed in 2–3 layers with 2-0 and 3-0 vicryl. Derma-bond, occlusive dressing and tape were placed over the wound. An abdominal binder was made available to the participant to increase comfort. A prescription for antibiotics was given for 5 days. Bandages were removed at the 7–10-day mark after implantation when recovery was determined to be complete. The participants’ incision sites were examined during the first month for wound checking. The participants were given instructions not to perform strenuous physical activities without immobilization following implantation for the first 3–4 weeks. Spinal mapping occurred during the fourth week of this process.

### 2.8. Migration of the Leads

Migration (displacement of the leads from the original anatomical placement following implantation) was determined as previously highlighted in our earlier reports [15,25]. Using image J software (version 1.54g), migration of the left and right leads was measured in participant 0884 after using images captured via X-ray fluoroscopy (following permanent implantation) or dual-energy X-ray absorptiometry (DXA; during follow-up visits). Due to ease of use, DXA scans were captured every three months as a screening tool. Measurements were analyzed to determine the placement of the leads relative to the original placement following permanent implantation in the cephalic–caudal direction or in the medio-lateral position. Medio-lateral migration refers to the distance between the right and left percutaneous leads.

### 2.9. Spinal Mapping

Spinal mapping was conducted daily following both temporary (1 week) and permanent (2 weeks) implantation, following 3–4 weeks of non-strenuous physical activity. Additionally, spinal remapping was conducted 3 months later for participant 0884 to determine the effects of migration on the mapping outcomes. In this process, various electrode configurations consisting of different anode/cathode configurations, pulse widths and stimulation frequencies were tested to achieve a multitude of functions without unwanted movements. Optimal electrode configurations were determined, including stimulation frequencies (2 Hz) and pulse durations (250, 500 or 1000 µs). The minimum amount of current (1–14 mA) was applied to evoke desirable motor activity in the target muscles. Wide-field and narrow-field configurations were selected to allow for standardization of the configurations tested to enable comparison between the participants. These wide-field and narrow-field configurations were shown to successfully achieve differential modulation of spinal motor pools in previous work [33]. Optimization of the configurations was highlighted in our previous work and was tailored to each participant’s unique anatomy based on an analysis of the magnitude of their individual extensor evoked potentials compared to flexors [20].

### 2.10. EMG Data Analysis

In the supine position, after shaving and cleaning the skin, ten EMG sensors were attached to the following muscles: the left and right vastus medialis (VM), the left and right rectus femoris (RF), the left and right hamstrings (HS), the left and right tibialis anterior (TA), and the left and right medial gastrocnemius (MG). The participant was asked to move the entire leg into flexion and extension, dorsiflex the ankle joint, and wiggle the big toe. At both temporary and permanent mapping, various electrode configurations with different stimulation parameters were tested to achieve optimal activation of the paralyzed leg muscles [15,20,23].

### 2.11. Temporary Versus Permanent Mapping

For the purpose of the current work, the configurations tested in temporary mapping and permanent mapping were retrospectively analyzed to identify matches in anode/cathode placement, stimulation frequency and pulse width to compare the recruitment curves (motor evoked potentials plotted versus stimulation amplitudes) of different muscles between the temporary and permanent mapping phases (Figure 3). For simplicity, left-leg motor evoked recruitment curves were compared after closely matching the stimulation programs between the temporary and permanent mappings.

Surface EMG activity was recorded for the left key muscles, including the rectus femoris (RF), the vastus medialis (VM), the hamstrings (HS), the tibialis anterior (TA) and the medial gastrocnemius (MG). All EMG signals were collected at a 2000 Hz sampling rate using LabChart 8.1.21 (Windows; A.D. Instruments, Sydney, Australia). The Root Mean Square (RMS) method was used to quantify the amplitude of muscle activation over a 5 s interval in response to each stimulation amplitude and for each stimulation configuration that was matched between temporary and permanent mappings [23]. Resting EMG activity was quantified at 5 s intervals without any stimulation for each channel and used to adjust baseline drift during temporary and permanent mappings. All temporary- and permanent-mapping motor evoked responses were normalized to the maximum value for each individual muscle across all configurations. Recruitment curves were developed to analyze the progression in muscle evoked potential amplitude in response to increasing stimulation amplitude [33]. Agreement between temporary and permanent mappings was defined as similar visual patterns in the curves. Visual agreement was then further assessed by plotting data points for temporary against permanent mappings or against remapping to calculate R^2^ values to examine the strength of the relationship between two different mapping sessions. Cohen’s D cut-off points were used to determine the strength of the agreement [small agreement = 0.2, medium agreement = 0.5 and large agreement = 0.8].

### 2.12. MRI Spinal Cord Reconstruction

Two-dimensional MRI scans of the spinal cord of each participant were recorded prior to SCES implantation (Figure 2 and Figure 4). The images were originally captured to ensure the patency of the lumbosacral region and to exclude any potential mass that may have precluded threading of the leads in the epidural space. For the purpose of the current work, the images were retrospectively analyzed to reconstruct the lumbosacral segments (L1-S2) relative to the original placements of the temporary and permanent leads. The images were converted to DICOM format, following which the DICOM browser application in MATLAB was used to view the images and isolate the required images. This was followed by a sagittal section of the lumbosacral region to identify the extent of the spinal cord. Subsequently, isolation of the axial scans was performed, and the images were processed in MATLAB and then loaded into the Volume segmented application to create a 3D reconstruction. The spinal cord was traced from the most caudal section to the most rostral rection. After the length of the spinal cord was identified, the spinal segments were marked, starting from S5 and moving caudally; previous studies in the literature were used to identify the average lengths of the spinal segments [34,35]. The model was extracted, and the representative images of spinal vertebrae were added to identify the location of the leads. Fluoroscopy and DXA images of the participants were analyzed using image J software (version 1.54g) to identify the location of the leads and to measure their length and placement. Within Image J, the scale for the measuring tool was set by measuring a known distance of 1 cm over the selected fluoroscopy image and setting up a conversion of pixels to centimeters. After the scale was set, measurements were taken for the leads, vertebrae and intervertebral discs. A difference in lead location between temporary and permanent implantation was determined by measuring the distance from the caudal end of the T12 vertebral body in each fluoroscopy image. Then, the leads at each implantation were overlayed on the reconstructed spinal cord to determine the proximity to the target region of the spinal cord (L1-S2 segments). These steps were retrospectively repeated for both participants to identify lead placements following temporary and permanent implantations.

## 3. Results

Figure 3
highlights spinal cord mappings following temporary and permanent implantation in five muscle groups (RF, VM, HS, TA and MG) for participant 0883. All configurations were tested for 250 µs, at a frequency of 2 Hz and at different amplitudes. There was relative agreement following wide-field and narrow-field caudal cathode configurations only in the VM muscle group. Agreement was also identified in the HS muscle group for the wide-field caudal cathode configuration. However, the rest of the configurations did not yield visual agreements between temporary and permanent mappings for the remaining muscle groups (Table 1). In subject 0883, visual agreement was supported by the resulting R^2^ values for the right leg (R^2^: 0.81; R^2^: 0.85; R^2^: 0.88) in each case, but not for the left leg (R^2^: 0.02; R^2^: 0.07; R^2^: 0.21) (Table 1).

The MRI spinal cord reconstruction presented in Figure 2 indicated that percutaneous leads were inadvertently placed differently between temporary and permanent implantations. During temporary mapping, leads covered primarily the L1 and most of the L2 segment (only three contacts per lead). However, permanent implantation covered only the L1 segment and the upper portion of the L2 segment (only three contacts per lead).

Figure 5 highlights spinal cord mapping following temporary and permanent implantation in five muscle groups (RF, VM, HS, TA and MG) for participant 0884. One wide-field configuration was only tested at 6 mA, with a frequency of 2 Hz and a pulse duration of 500 µs, for temporary mapping and repeated at 6 and 8 mA during permanent mapping. An amplitude of 8 mA was added during permanent mapping to determine whether the magnitude of the motor evoked response would vary or remain unchanged due to the staggered nature of the configuration. The results indicated relative visual agreements in four of the five muscle groups, except for the HS muscle (Table 1). It is interesting to note that the HS muscle showed a mirror representation of the motor evoked response between the left and right leg following temporary and permanent implantation. Figure 5 highlight the spinal cord reconstruction following temporary and permanent implantation in participant 0884. Unlike participant 0883, temporary (six contacts per lead) and permanent (six contacts per lead) implantation covered the mid-L3 down to the S2 segment, which may explain the improvement in the relative agreement of the motor evoked response between the temporary and permanent mappings. In subject 0884, R^2^ values were not determined between temporary and permanent mappings because only one data point was tested (Table 1 and Figure 5).

### 3.1. Percutaneous Lead Migration in Participant 0884

Following permanent mappings, medial migration (displacement of the leads from the original anatomical placement following implantation) of the top contact in the left lead occurred in participant 0884 (Figure 6, Figure 7 and Figure 8). A DXA scan was captured approximately 3 months after implantation to measure migration relative to the fluoroscopy that was captured on the day of permanent implantation (Figure 6). Figure 6 highlights the migration in percutaneous leads between the permanent implantation and three months later.

Following permanent implantation (A), the distal ends of the left and right leads were placed evenly at 1.8 cm from T12. The distance from the upper rim of the T11 vertebra to the proximal tip of both leads was 0.62 cm. To account for mediolateral migration (displacement of the leads from the original anatomical placement following implantation), the distance between proximal contacts 3 and 11 was 0.68 cm, and the distance between contacts 7 and 15 was 0.65 cm. Three months later, the left lead migrated proximally, and the distance was 2 cm from T12, and the right lead remained unchanged. The distance from the upper rim of the T11 vertebra to the proximal tip of the leads was 0.59 cm for the right lead and 0.73 cm for the left lead. To account for mediolateral migration, the distance between proximal contacts 3 and 11 shrunk to 0.2 cm, and the distance between contacts 7 and 15 was 0.62 cm. Three months later, there was crossover of both leads at the contacts of 0 and 8 and 1 and 9.

### 3.2. Remapping of the Spinal Cord Following Implantation

Remapping was conducted in two wide-field and one-narrow field configuration. All configurations were tested at three pulse durations (250, 500 and 1000 µs). Despite migration, the relative agreement across the three configurations at different pulse durations was high (Table 2). However, migration attenuated the response of the RF and HS muscle groups in the remapping stage, especially in the two-wide field configurations (Figure 7 and Figure 8). R^2^ values were able to strongly support the finding of visual agreement in most cases between mapping periods after migration (Table 2).

## 4. Discussion

The primary goal of the current work was to explore the utility of temporary spinal mapping to guide the permanent implantation of SCES leads and identify configurations for promoting motor recovery after SCI. Our results suggested that temporary mapping may potentially be capable of guiding the placement of the leads, assuming the availability of spinal cord reconstruction of the lumbosacral segments (L1-S2). MRI reconstruction of the spinal cord suggested that the L1-S2 anatomical distribution varied between participants 0883 and 0884. Despite efforts to implants the leads to cover the T11 and T12 vertebrae, the L1 segment corresponded to the mid-T12 vertebra in both participants. This led to higher placement of the leads in 0883, with only the bottom three contacts per single lead covering the L1 and L2 segments. The segments from L3 to S2 were not covered by any active cathodes, which may explain why participant 0883 could not achieve functional standing [20]. Migration clearly attenuated the motor evoked potentials of the RF and HS muscle groups and resulted in the loss of functional standing in participant 0884. This may have potentially resulted in an alteration to the synergistic motor activities between these two agonist–antagonist muscle groups. However, the relative agreements were high in the other muscle groups that were not impacted by migration of the leads (Table 2) during the mapping and remapping sessions (Figure 6).

Unlike other centers [11,12,13,14], intraoperative mapping was not an option in our center during permanent implantation [15]. We attempted intraoperative mapping during temporary implantation by moving the percutaneous leads using the fluoroscopy guidance in the region of the T11-L1 vertebrae. However, there were faded motor evoked potentials of the tested muscles, and noticeable or visible muscle contractions were barely elicited. It is possible that the process of lead steering over four to five vertebral body lengths or an air/saline infusion opening the epidural space may have transiently deactivated certain cord segments, making them unrecruitable. A few hours later, motor evoked potentials were clearly elicited following temporary implantation on the same day. It is well established that temporary mapping serves an important step in applications of percutaneous SCES for analgesic purposes [30]. During a trial mapping period, percutaneous leads are threaded into the epidural space and connected to an external neurostimulator, and tonic stimulation is optimized. Based on the reduction in pain intensity, a decision is made on whether to proceed to permanent implantation or to exclude the participant from the trial.

In the current trial, temporary implantation was originally utilized to determine the comfort level of each participant and to determine whether or not we could actively stimulate the target paralyzed muscles based on specific configurations. This approach would allow patients improved informed consent for an upcoming implantation while allowing teams to safely disqualify participants in case of the development of autonomic dysreflexia, uncontrolled spasms or spasticity after temporary implantation.

The duration of temporary implantation was only 5 days and allowed us to test specific SCES configurations that were found to be most reliable upon testing in the first 2–3 days [15]. After the third day, migration of the percutaneous leads was likely to conflict with the mapping results because of the external nature of the stimulator that was glued on the back of each participant as well as the movement in and out of the wheelchair [15]. Therefore, every effort was considered to finalize temporary mapping in the first 3 days following implantation. Previously, others used intraoperative mapping to determine the accurate placement of the paddle [12,13]. Hofstoetter et al. identified a statistical association between intraoperative and postoperative mapping of the spinal cord [24]. The authors found that intraoperative mapping predicted motoneuron pool activation obtained in postoperative mapping [24]. Therefore, potentially proposing to utilize temporary mapping may guide future implantation in a similar fashion to intraoperative mapping.

Our preliminary work demonstrated that percutaneous SCES may lead to the achievement of trunk control, overground standing and the restoration of overground stepping in a non-functional pattern [15,20,25]. Two out of the four participants restored standing after successfully modulating the extensor-to-flexor muscle activities necessary to maintain hip, knee and ankle extension [20]. However, the reasons for the lack of independent standing in the other two participants were unclear. Similar to an earlier report [26], we successfully reconstructed the lumbosacral segments of the spinal cord using the captured axial MRI images. The spinal cord reconstruction provided insights into the anatomical locations of the percutaneous leads relative to the L1-S2 segments. In participant 0883, 62% and 75% of the contacts did not anatomically correspond to the target segments during temporary and permanent implantation, respectively. For 0884, 63% and 75% of the contacts successfully targeted the lumbosacral segment from L3 to S2. This may explain why participant 0884 managed to achieve partial standing initially after permanent implantation (despite improper coverage of L1-L2).

The current findings highlighted the significance of utilizing MRI to determine the end of the conus of the cord. It is typically assumed that the spinal cord ends between the L1/L2 intervertebral space. However, the spinal cord of participant 0883 terminated between the L2/L3 intervertebral space, and for 0884, the spinal cord terminated in the lower third of the T12 vertebra. This confirms previous results that indicated wider variability in the length of the spinal cord, especially in persons with SCI [7,26,29]. For both participants, approximately 1.5 cm separated the S5 segment from the S2 segment. This may provide clinical guidance in future implantations, where the leads may be threaded to be placed approximately 1.5 cm from the conus end of the spinal cord. Therefore, MRI reconstruction of the cord is essential to ensure target stimulation of the lumbosacral segments in persons with SCI.

Previous reports utilized MRI to predict future walking and motor recovery in persons with SCI [36,37]. Angeli et al. noted the importance of using a 3D reconstruction model of the spinal cord and recording intraoperative potentials to guide paddle placement [19]. MRI reconstruction of the lumbosacral segment also explained the differences in motor evoked potential between temporary and permanent mappings. For 0883, motor evoked potentials were clearly recognized for the VM/HS, RF/HS and MG/TA muscle pairs across different configurations following temporary implantation. We applied both wide-field and narrow-field configurations; however, 75% of the contacts were not in the target zone and resulted in non-specific stimulation of the target segments, except at higher amplitudes (9–10 mA). It is worth noting that pulse duration was also limited to 250 µs in participant 0883. The choice of 250 µs was retrospectively considered to allow matching and comparisons between temporary and permanent configurations in this participant. The upward placement of the leads attenuated the motor evoked potentials of the VM/HS, RF/HS and MG/TA muscles across different configurations and resulted in non-specific stimulation, especially following permanent implantation.

On the contrary, greater evoked potentials were noted in the three muscle pairs in participant 0884 during permanent mapping. Permanent implantation resulted in 75% of the contacts targeting the L3-S2 segments compared to only 67% following temporary implantation. Previous researchers developed an anatomical model of the spine and spinal cord to map specific motor pools via segmental stimulation and altering cathodal placements [26]. Hofstoetter et al. used anatomical parameters from thousands of individuals to create a statistical model of the spinal cord [24]. This model provided a relationship between the stimulation site and the activated spinal cord segments [24]. The recruitment of the leg muscles in temporary mapping may not be replicated in permanent mapping without prior knowledge of the exact anatomical placement of the leads with respect to specific spinal cord segments.

Migration following percutaneous implantation was previously reported in persons with SCI. Our results agree with our initial report that showed lead migration in both the cephalic–caudal and medio-lateral directions. A previous report indicated that 88.5% of implanted leads showed caudal migration within 20 days following implantation. Participant 0884 achieved functional standing by intentionally locking their hips and knees against gravity without external support and by using a standard walker [20]. Early activity in the form of vibrational impact using a riding mower prior to the 4-week period of rest, intended to promote epidural scarring, may have contributed to the rare presentation of medial lead migration. Most spinal cord lead migration is caudal [38]. Spinal cord remapping indicated attenuated response in the motor activity of the RF and HS muscle groups, especially in the left leg. Previous work indicated the synergistic role of RF and HS muscle to achieve knee extension and enhanced hip extension [39]. Therefore, a reduction in RF activities following migration may have resulted in an increase in the flexion activity of the HS muscle group and failure to ensure functional standing.

### Limitations

Several limitations need to be acknowledged regarding the current work. The first point is that the data were retrospectively analyzed, so perfect matching between the temporary and permanent mappings was not always possible, especially for participant 0884. Temporary mapping was conducted on the following day compared to permanent mapping, which was conducted 3–4 weeks later. It is typical to assume that temporary implantation may reflexively result in spinal inhibition that impacts the outcomes of motor evoked response. However, this was not the case in participant 0883. Furthermore, limited pulse durations were examined in participant 0883 (short pulse duration: 250 µs) compared to a longer pulse duration in 0884 participant (500 µs) during temporary and permanent mapping sessions. A shorter pulse duration may have further exacerbated the non-target-specific stimulation in participant 0883. However, this limitation was addressed when spinal cord mapping and remapping were performed in participant 0884 by examining different pulse durations (250, 500 and 1000 µs). The MRI data were analyzed retrospectively, and knowledge about the conus end would have better guided the implantation in participants 0883 and 0884. Finally, the small sample size (n = 2) may limit the generalizability of the current findings to other participants who may likely undergo implantation in the future. It is important to note that the SCI population is highly heterogeneous, with differences in length of time since injury, health status and spinal cord anatomy. Due to these differences, temporary mapping with MRI reconstruction may not necessarily be as effective for guiding the permanent implantation of leads for all individuals. The two participants’ unique attributes, such as chronicity of injury, age and differences in rehabilitation protocols, may have served as potential confounding variables, influencing the results. With regard to MRI reconstruction, it is imperative that MRI in future studies be performed for the purpose of collecting data for spinal cord reconstruction, with thinner slice sections, higher resolutions and no gaps in between slices to provide more accurate and detailed images to facilitate reconstruction. The reconstructed lumbosacral spinal cord segments may not be accurate anatomical representations for both participants because the images were not collected initially for this purpose. We primarily relied on existing images as well as published data to feed into our reconstruction model. Possibly, the thickness of spinal cord segments reported in the literature may vary between individuals, impacting the accuracy of the developed model [35,36]. Due to these shortcomings, it is important to note that the findings presented here should be considered as preliminary and to highlight the need for further investigation into MRI-based mapping. Furthermore, the translation aspect of the current work will be enhanced by validating the current findings with a larger sample size.

## 5. Conclusions

We attempted to identify the agreement between temporary and permanent mapping results in two individuals with SCI. During both mapping periods, various electrode configurations with different stimulation parameters were tested to achieve the optimal segmental activation of the paralyzed leg muscles. Our results showed a discrepancy in the recruitment of the leg muscles between the temporary and permanent mappings, which was linked to the placement of the leads, as demonstrated by MRI reconstruction. Our results, therefore, suggest that temporary mapping with MRI reconstruction of the spinal cord may be used to achieve optimal activation of the spinal motor pools and guide permanent mapping during percutaneous SCES implantation. Knowledge about the MRI location of the conus end is essential to determine the exact location for anatomical placement of the leads to target L1-S2 segments. Our results suggested that placing the caudal end of the leads 1.5 cm above the end of the conus would accurately align the leads against the S2 spinal cord segment. Migration modestly dampened the activity of synergistic muscle groups and resulted in loss of functional standing in a person with SCI. Future strategies should consider using evolved and stronger implanted anchors to reduce migration of percutaneous leads in persons with SCI, similar to interventional pain management trials [40,41]. Medial migration, although less common, may have occurred before the leads had adequate time to form epidural scars. Therefore, it is important to ensure an adequate recovery period, possibly 4 weeks, following permanent implantation.

## Figures and Tables

**Figure 1 jcm-13-06826-f001:**
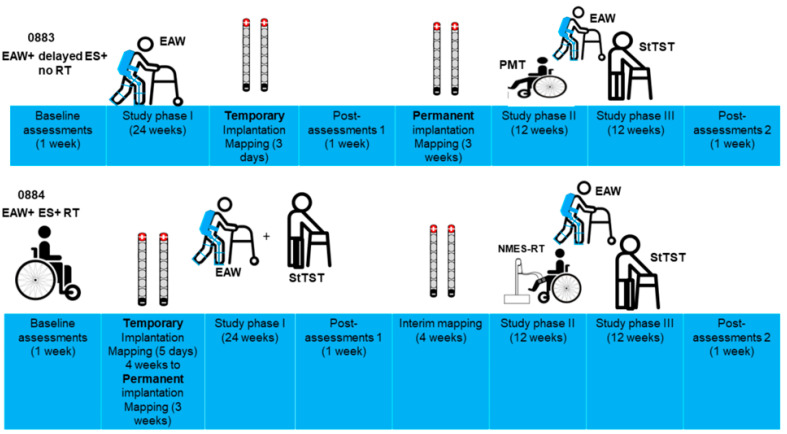
Timeline of study phases for the two participants, 0883 and 0884. Both participants were randomized into either an exoskeletal-assisted walking (EAW) + 6 months delayed SCES + no resistance training (RT) group or EAW + SCES + RT. The timeline of the interventions was approximately 12 months separated by measurements at the 6-month period at baseline, post-assessment 1 and post-assessment 2 (1 week each; results are not reported). After baseline measurement, temporary implantation was performed, followed by 3–5 days of mapping. Following post-assessment 1, permanent implantation was performed and followed with mapping for another 3 weeks. The first 12 weeks following implantation included passive movement training (PMT; twice weekly) to balance the design with the other group (EAW + SCES + RT). For participant 0884, phase I of the study began with 6 months of training 3 days per week. After 24 weeks, one week of reassessing outcome measures (post-assessment 1) and four weeks of remapping were conducted to optimize standing and walking functions for the next phase of the study (interim mapping). Phase II of the study consisted of 12 weeks of the same training as in phase I, plus 2 days/week of neuromuscular electrical stimulation resistance training (NMES-RT) on alternate days to SCES + EAW and SCES + StTST. This was followed directly by phase III of the study, in which the participants continued the same training as in phase I but where NMES-RT was replaced by SCES + sit-to-stand practice with a walker for 12 weeks. In the final week, the outcome measures were reassessed (post-assessment 2).

**Figure 2 jcm-13-06826-f002:**
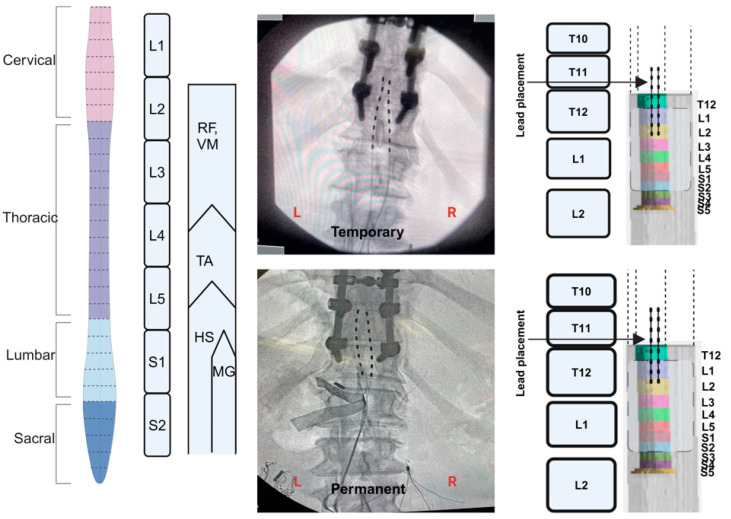
Spinal cord model highlighting the lumbosacral innervation of the corresponding muscles. Fluoroscopy images following the conclusion of temporary and parament implantation in subject 0883. Retrospective 3D MRI reconstruction of the spinal cord and estimated placement of the percutaneous SCES leads during temporary and permanent mapping (subject 0883). The reconstructed spinal cord was based on axial MRI of the lumbosacral segments prior to implantation. The lower 3 contacts per lead covered the L1 and L2 segments. The measured distance from S5 to S2 was 1.5 cm. (Created in BioRender; Rehman, M. (2024); BioRender.com/d70p619 (accessed on 6 November 2024).).

**Figure 3 jcm-13-06826-f003:**
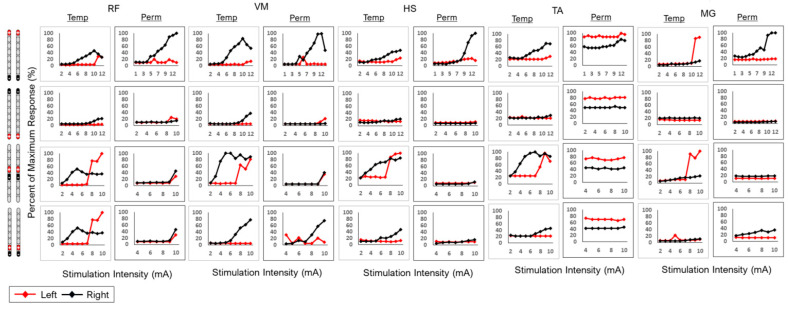
Agreement between temporary and permanent mappings for different muscle groups (RF, VM, HS, TA and MG) in subject 0883. Four configurations were tested with two wide-field and two narrow-field configurations. All configurations were tested at 2 Hz with a pulse duration of 250 µs. Visual agreement was used to determine the similarity between the recruitment curves of the right-side temporary and right-side permanent or the left-side temporary and left-side permanent mappings.

**Figure 4 jcm-13-06826-f004:**
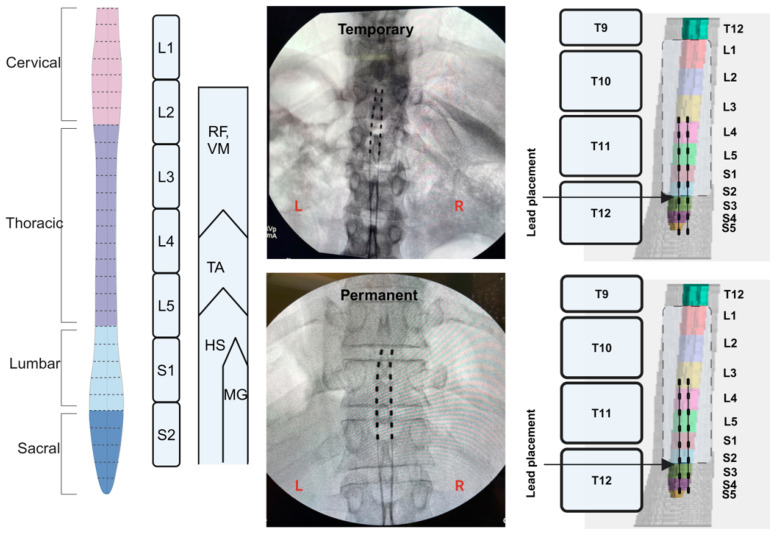
Spinal cord model highlighting the lumbosacral innervation of the corresponding muscles. Fluoroscopy images following the conclusion of temporary and parament implantation in subject 0884. Retrospective 3D MRI reconstruction of the spinal cord and estimated placement of the percutaneous SCES leads during temporary and permanent mapping (subject 0884). The reconstructed spinal cord was based on axial MRI of the lumbosacral segments prior to implantation. The upper 6 contacts per lead covered the L3 and S2 segments. The measured distance from S5 to S2 was 1.5 cm. (Created in BioRender; Rehman, M. (2024); BioRender.com/w78b669 (accessed on 6 November 2024).).

**Figure 5 jcm-13-06826-f005:**
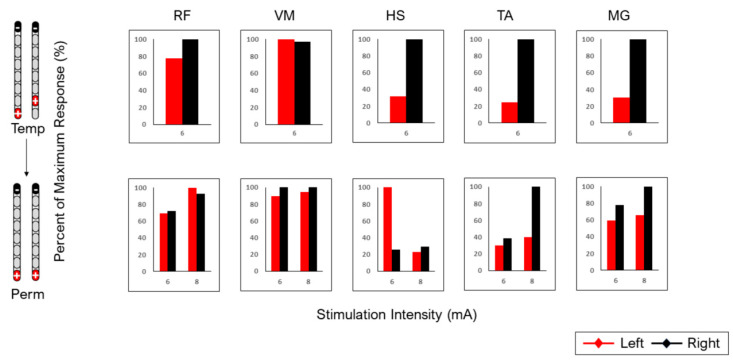
Agreement between temporary and permanent mappings for different muscle groups (RF, VM, HS, TA and MG) in subject 0884. Only one amplitude (6 mA) was tested at a frequency of 2 Hz with a pulse duration of 500 µs during temporary implantation. Motor evoked responses were attenuated or remained silent at amplitudes of 1–4 mA following temporary implantation. Because of the retrospective nature of the work, amplitudes of 6 and 8 mA were chosen at a frequency of 2 Hz with a pulse duration of 500 µs during permanent mapping to determine the visual agreement compared to temporary mapping. The 8 mA amplitude was included because of the difference in anodal locations between temporary (+14) and permanent (+15) mappings of the chosen staggered configuration, which may have accounted for different motor evoked responses. The temporary and permanent motor evoked responses were similar for 4 muscle groups (RF, VM, TA and MG) and showed high visual agreement, except for HS. The HS muscle group showed a mirror response of the motor evoked potentials between the left and right sides when comparing permanent responses to temporary responses.

**Figure 6 jcm-13-06826-f006:**
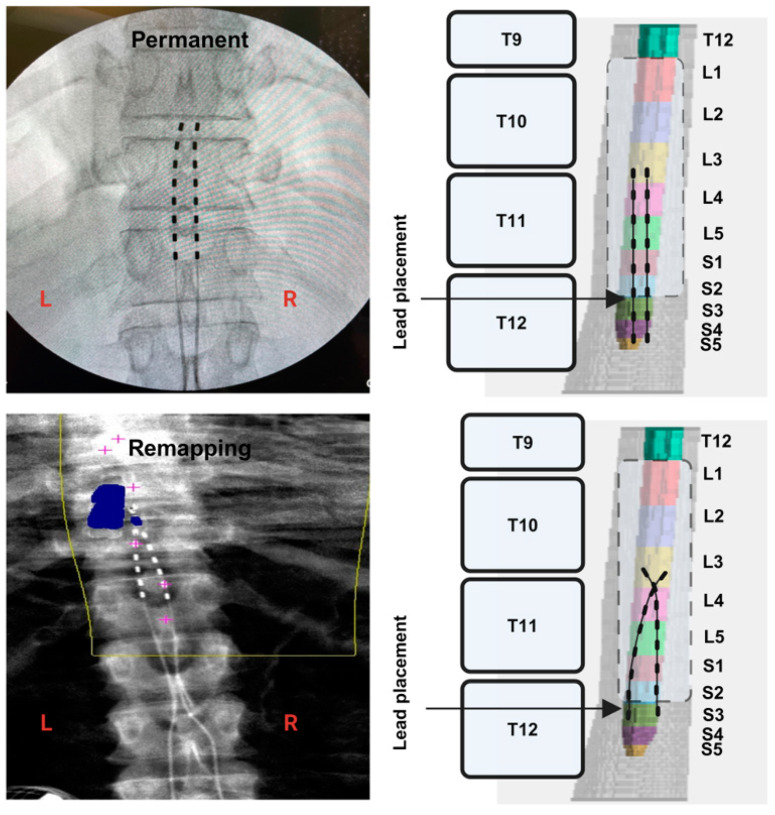
Lead migration in participant 0884. The tip of the left lead crossed over to the right lead at the L3 segment. Fluoroscopy images following the conclusion of permanent implantation in subject 0884. Lead migration was detected by dual-energy X-ray absorptiometry of the spine region. (Created in BioRender; Rehman, M. (2024); BioRender.com/t49c027 (accessed on 6 November 2024).).

**Figure 7 jcm-13-06826-f007:**
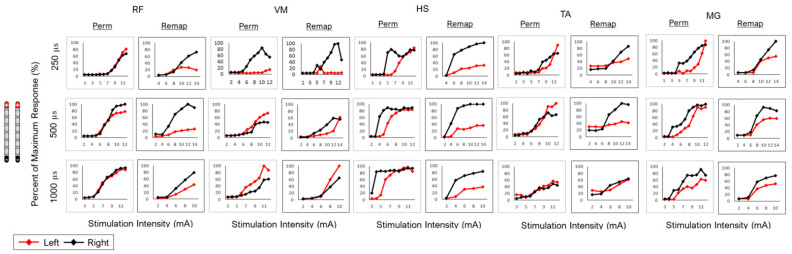
Wide-field configuration with caudal cathode placement in subject 0884 following permanent implantation (perm mapping) and migration of the leads (remap: remapping) at different pulse durations (250, 500 and 1000 µs) in different muscle groups (RF, VM, HS, TA and MG). Visual agreement indicated attenuated responses in the left RF and HS following migration.

**Figure 8 jcm-13-06826-f008:**
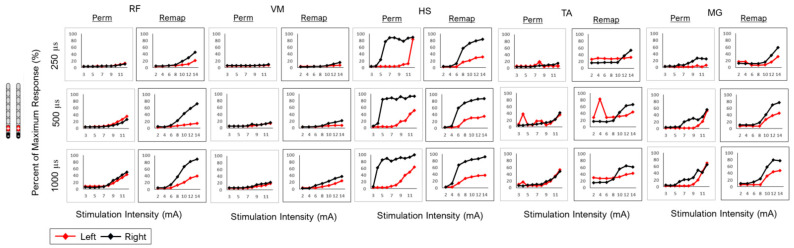
Caudal narrow-field configurations in subject 0884 following permanent implantation (perm mapping) and migration of the leads (remap: remapping) at different pulse durations (250, 500 and 1000 µs) in different muscle groups (RF, VM, HS, TA and MG). Visual agreement indicated attenuated responses in the left RF and HS following migration. The current had to be increased to 14 mA to achieve a similar response to what was achieved during permanent implantation. Results are listed in Table 2.

**Table 1 jcm-13-06826-t001:** Agreement between temporary and permanent mappings in two persons with SCI.

Configuration	Muscle	Agreement Between Temporary and Permanent Spinal Mappings
Subject 0883		
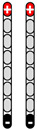	RF	X
	VM	A (R^2^: 0.02, 0.81)
	HS	A (R^2^: 0.07, 0.85)
	TA	X
	MG	X
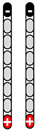	RF	NR
	VM	NR
	HS	NR
	TA	X
	MG	NR
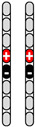	RF	X
	VM	X
	HS	X
	TA	X
	MG	NR
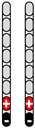	RF	X
	VM	A (R^2^: 0.21, 0.88)
	HS	X
	TA	X
	MG	X
Subject 0884		
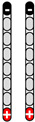	RF	A
	VM	A
	HS	X
	TA	A
	MG	A

A: visual agreement between temporary and permanent mapping sessions; X: no agreement found; NR: no response as a result of different configurations. R^2^ values included where applicable, with the left leg presented first, followed by the right leg.

**Table 2 jcm-13-06826-t002:** Agreement between mapping and remapping following migration in a person with SCI.

Configuration	Muscle	Agreement Between Permanent and Spinal Remapping
Subject 0884		
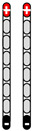	RF	A [250 µs: (R^2^: 0.75, 0.99); 500 µs: (R^2^: 0.99, 0.95); 1000 µs: (R^2^: 0.98, 0.98)]
	VM	A [250 µs: (R^2^: 0.99, 0.94); 500 µs: (R^2^: 0.95, 0.91); 1000 µs: (R^2^: 0.97, 0.99)]
	HS	X
	TA	A [250 µs: (R^2^: 0.76, 0.74); 500 µs: (R^2^: 0.88, 0.94); 1000 µs: (R^2^: 0.99, 0.75)]
	MG	A [250 µs; (R^2^: 0.77, 0.79); 500 µs: (R^2^: 0.93, 0.93); 1000 µs: (R^2^: 0.95, 0.86)]
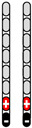	RF	X
	VM	A [250 µs: (R^2^: 0.99, 0.99); 500 µs: (R^2^: 0.94, 0.73); 1000 µs: (R^2^: 0.98, 0.92)]
	HS	A [250 µs: (R^2^: 0.55, 0.52); 500 µs: (R^2^: 0.56, 0.93); 1000 µs: (R^2^: 0.66, 0.91)]
	TA	X
	MG	X
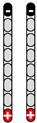	RF	X
	VM	X
	HS	X
	TA	X
	MG	X

A: visual agreement between temporary and permanent mapping sessions; X: no agreement found; NR: no response as a result of different configurations. R^2^ values included where applicable, with the left leg presented first, followed by the right leg. Multiple pulse durations were also specified and tested between mapping and remapping for 0884.

## Data Availability

All data reported in this paper will be shared by the lead contact author (Ashraf Gorgey (ashraf.gorgey@va.gov)) upon request and after obtaining necessary approval from local research office.

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
