# Peer review of "MRI Spinal Cord Reconstruction Provides Insights into Mapping and Migration Following Percutaneous Epidural Stimulation Implantation in Spinal Cord Injury"

_jcm, 2024, doi:10.3390/jcm13226826_

Round 1

Reviewer 1 Report

Comments and Suggestions for Authors

The original article by Venigalla, et al."MRI-Spinal Cord Reconstruction Provides Insights on Mapping and Migration following Percutaneous Epidural Stimulation Implantation in Spinal Cord Injury" covers a potentially interesting and emerging topic related to the SCI therapy. In this sense, this remains to be potentially interesting for the JCM readers. I regard the main point of this paper as highly attractive as well as the results are clearly presented. The text does not contain any major errors, therefore I have some minor comments and recommendations:

1. There is a need to provide slightly more expanded introduction shortly mentioning/describing pathogenesis of SCI and its impact of modern healthcare.

2.   The figure summarizing and clarifying the results should be added.

3. Following references should be added and properly cited within the main text to improve the quality of manuscript:

- Turczyn P, Wojdasiewicz P, Poniatowski ŁA, Purrahman D, Maślińska M, Żurek G, Romanowska-Próchnicka K, Żuk B, Kwiatkowska B, Piechowski-Jóźwiak B, Szukiewicz D. Omega-3 fatty acids in the treatment of spinal cord injury: untapped potential for therapeutic intervention? Mol Biol Rep. 2022 Nov;49(11):10797-10809. doi: 10.1007/s11033-022-07762-x.

- Huang Q, Duan W, Sivanesan E, Liu S, Yang F, Chen Z, Ford NC, Chen X, Guan Y. Spinal Cord Stimulation for Pain Treatment After Spinal Cord Injury. Neurosci Bull. 2019 Jun;35(3):527-539. doi: 10.1007/s12264-018-0320-9. Epub 2018 Dec 17. PMID: 30560438; PMCID: PMC6527651.

4. In some places the use of English throughout the whole manuscript could be improved on.

Completing this gaps will have an impact on the understanding the aim of the study and, from my point of view, is absolutely necessary.

Comments on the Quality of English Language

minor review

Author Response

Reviewer 1

The original article by Venigalla, et al."MRI-Spinal Cord Reconstruction Provides Insights on Mapping and Migration following Percutaneous Epidural Stimulation Implantation in Spinal Cord Injury" covers a potentially interesting and emerging topic related to the SCI therapy. In this sense, this remains to be potentially interesting for the JCM readers. I regard the main point of this paper as highly attractive as well as the results are clearly presented. The text does not contain any major errors, therefore I have some minor comments and recommendations:

  1. There is a need to provide slightly more expanded introduction shortly mentioning/describing pathogenesis of SCI and its impact of modern healthcare.

Thank you to the reviewer for this valuable feedback. We agree that it would be valuable to provide information about the pathogenesis of SCI and its healthcare impact to the introduction. As requested, we have added the following lines:

“Spinal cord injury (SCI) is a debilitating neurological state primarily caused by vertebral fracture or dislocation leading to neuronal damage [Anjum et. al, 2020]. What ensues is glial scarring and inflammation at the site of injury [Ming-Ping et. al, 2021]. SCI is associated with a range of comorbidities including limited functional mobility, respiratory dysfunction, cardiovascular dysfunction, depression, sexual dysfunction and bladder and bowel issues [Guest et. al, 2022]. SCI also poses a significant financial burden on healthcare with one study estimating the lifetime expenditure per individual with SCI to be $0.7- 2.5 million [Diop and Epstein, 2024].”

  1. The figure summarizing and clarifying the results should be added.

Thank you to the reviewer for this suggestion. While we agree concise results and figures are vital, we believe we have provided figures that summarize results in the current version of the manuscript (Figures 1, 2 and 4). We would like to point the reviewer to Table 1 which summarizes the agreement in the motor evoked potentials for both participants for each muscle group. We believe this table offers a concise and summarized version of the motor recruitment curve results presented in the other figures. Figures 2 and 4 also provide a central overview of the lead placement in 0883 and 0884 with respect to the spine and spinal cord during temporary and permanent mapping.

  1. Following references should be added and properly cited within the main text to improve the quality of manuscript:

- Turczyn P, Wojdasiewicz P, Poniatowski ŁA, Purrahman D, Maślińska M, Żurek G, Romanowska-Próchnicka K, Żuk B, Kwiatkowska B, Piechowski-Jóźwiak B, Szukiewicz D. Omega-3 fatty acids in the treatment of spinal cord injury: untapped potential for therapeutic intervention? Mol Biol Rep. 2022 Nov;49(11):10797-10809. doi: 10.1007/s11033-022-07762-x.

- Huang Q, Duan W, Sivanesan E, Liu S, Yang F, Chen Z, Ford NC, Chen X, Guan Y. Spinal Cord Stimulation for Pain Treatment After Spinal Cord Injury. Neurosci Bull. 2019 Jun;35(3):527-539. doi: 10.1007/s12264-018-0320-9. Epub 2018 Dec 17. PMID: 30560438; PMCID: PMC6527651.

Thank you to the reviewer for suggesting these references. We believe these are excellent references to add and have inserted them into the main text. The following lines includes these references:

“Spinal cord injury is a debilitating neurological state primarily caused by vertebral fracture or dislocation leading to neuronal damage [Anjum et. al, 2020]. What ensues is glial scarring and inflammation at the site of injury [Ming-Ping et. al, 2021]. Spinal cord injury is associated with a range of comorbidities including limited functional mobility, respiratory dysfunction, cardiovascular dysfunction, depression, sexual dysfunction and bladder and bowel issues [Guest et. al, 2022]. Several interventions have been developed to treat spinal cord injury such as the use omega-3 fatty acids which have shown potential for their anti-inflammatory and neurodegenerative effects but require further validation in multicenter studies [Turczyn et. al, 2022]. Other interventions include the use of spinal cord stimulation to help treat injury-associated chronic pain which is promising but also requires further evidence on its effectiveness from future studies [Huang et. al, 2019].”

  1. In some places the use of English throughout the whole manuscript could be improved on.

Thank you for pointing this out. We have adapted the English throughout the manuscript in the following lines to enhance the clarity….

Completing this gaps will have an impact on the understanding the aim of the study and, from my point of view, is absolutely necessary.

Thank you to the reviewer for these insightful recommendations. We have implemented all your recommendations as requested.

Reviewer 2 Report

Comments and Suggestions for Authors

General comments:

This article addresses an important topic within the field of spinal cord injury (SCI) treatment, specifically the use of spinal cord epidural stimulation (SCES) and MRI-guided spinal cord reconstruction to improve motor function outcomes. The authors conducted a study involving two participants with clinically complete SCI to evaluate the utility of temporary SCES mapping in guiding permanent lead placement and compared motor evoked potentials (MEPs) between temporary and permanent implantations. This work contributes to the growing body of literature exploring non-invasive neuromodulation techniques and offers valuable insights into lead migration and motor function recovery.

The article is well-structured and follows a logical progression from the background to the discussion of results. The use of MRI reconstruction to map spinal cord segments and evaluate lead placement is a novel and potentially impactful approach. However, several critical methodological and conceptual issues need to be addressed before this manuscript can be accepted for publication. Moreover, several technical inaccuracies, assumptions, and limitations are not sufficiently acknowledged.

Major revisions:

1. The study's sample size (n=2) is a significant limitation. The article does not sufficiently address the limitations in generalizability, especially given that only two participants were studied. SCI patients are highly heterogeneous, and conclusions drawn from such a small cohort should be tempered. The authors need to explicitly discuss how the findings from these two participants may or may not translate to the broader population of SCI patients. A deeper consideration of variability between individuals with SCI, including differences in injury chronicity, anatomical variations, and overall health status, is essential to contextualize the findings.

  1. There is no control group for comparison, either a group receiving intraoperative mapping or a cohort using SCES without MRI guidance. While it may not be feasible to include such a group in this pilot study, the absence of a control leaves the findings relatively isolated. The authors should provide a more detailed justification for their study design choices and discuss potential confounding variables (e.g., participant age, injury chronicity, differences in rehabilitation protocols) that could affect the study’s outcomes.
  2. The interpretation of lead migration and its effect on motor function is somewhat inconsistent. The authors suggest that minor migration does not substantially affect motor function in some muscle groups, but this interpretation requires more careful analysis. The study reports "high agreement" in certain motor groups despite migration, but this is not statistically supported. The authors must clarify whether the analysis was qualitative or quantitative, and if quantitative, more rigorous statistical methods should be applied to test for significant differences. If the comparison is qualitative, this should be clearly stated and discussed as a limitation.
  3. While the authors present MRI reconstruction as a reliable tool for SCES lead placement, the method is not fully validated. The current retrospective approach could be prone to errors in segmentation and alignment. The authors should provide more technical details on how the accuracy of MRI-based reconstruction was verified and discuss potential sources of error in mapping lead locations. Additionally, the claim that MRI can replace intraoperative mapping is premature without larger validation studies. The manuscript should present these findings as preliminary and call for more extensive research to confirm the efficacy of MRI-based mapping.
  4. The study does not provide a clear analysis of the motor function improvements following permanent implantation. The reported outcomes focus on MEPs rather than functional outcomes like standing or stepping. While MEPs are important, the ultimate clinical goal is improved function, and the manuscript would benefit from a more detailed discussion of how the observed changes in MEPs translated (or did not translate) into meaningful improvements in motor function. The paper would also benefit from a comparison with previous studies using SCES for motor function recovery.

Minor revisions:

  1. The statistical methods used in this study are not well described. A more detailed explanation of how the data were analyzed, especially for agreement between temporary and permanent mapping, is necessary. If visual agreement was used as the primary measure, the limitations of this approach should be acknowledged, and where applicable, quantitative methods should be employed.
  2. The paper describes the use of wide-field and narrow-field electrode configurations but does not provide enough information on how these configurations were selected or optimized for each participant. More details on the rationale for configuration choices and how they were tailored to each participant’s anatomy would strengthen the methods section.
  3. Throughout the manuscript, there is some inconsistency in the use of technical terms (e.g., "motor evoked potentials," "recruitment curves," "migration"). The authors should ensure that these terms are consistently defined and used throughout the manuscript.
  4. Some of the references, particularly those related to MRI reconstruction and SCES implantation, are outdated. The authors should ensure that the most current literature is cited, particularly in the discussion of SCES technologies.

Language and grammatical revisions:

  1. In the abstract, "enhance motor function to achieve standing and stepping after SCI" is awkwardly phrased. It would be more precise to say "improve motor function, aiming to restore standing and stepping abilities in individuals with SCI."
  2. The phrase "Minor lead migration did not seem to impact the outcomes of spinal cord mapping" could be rephrased for clarity. Consider: "Minor lead migration appeared to have minimal impact on spinal cord mapping outcomes."
  3. In the introduction, "Research studies agreed that spinal cord mapping is an essential cornerstone step" is redundant. The phrase "essential cornerstone" should be revised for clarity, as "cornerstone" alone would suffice.
  4. In several sections, sentence construction is overly complex. For example, "We have chosen 8 mA because of difference in anodal locations between temporary (+14) and permanent (+15) mapping, which may have resulted in different motor evoked response" can be simplified for readability.
  5. The discussion section occasionally lapses into passive voice, which weakens the directness of the analysis. For example, "It was unclear why the other two participants did not achieve independent standing" could be revised to "The reasons for the lack of independent standing in the other two participants were unclear."

Recommendations:

This paper requires major revisions before it can be considered for publication. The primary issues revolve around the study design, interpretation of findings, and methodological limitations that should be addressed. Once these concerns are adequately resolved, the manuscript has the potential to make a meaningful contribution to the literature on SCES and motor recovery in SCI patients.

Comments on the Quality of English Language

The English language quality is generally adequate but requires improvement for clarity and precision. Some sentences are awkwardly constructed, and the passive voice is overused, which weakens the impact. Consistency in technical terminology should be ensured. A language revision to enhance readability and coherence is recommended.

Author Response

Reviewer 2

This article addresses an important topic within the field of spinal cord injury (SCI) treatment, specifically the use of spinal cord epidural stimulation (SCES) and MRI-guided spinal cord reconstruction to improve motor function outcomes. The authors conducted a study involving two participants with clinically complete SCI to evaluate the utility of temporary SCES mapping in guiding permanent lead placement and compared motor evoked potentials (MEPs) between temporary and permanent implantations. This work contributes to the growing body of literature exploring non-invasive neuromodulation techniques and offers valuable insights into lead migration and motor function recovery.

Thank you to the reviewer for their kind feedback. We totally appreciate your time and effort providing critical and insightful comments about the work.

The article is well-structured and follows a logical progression from the background to the discussion of results. The use of MRI reconstruction to map spinal cord segments and evaluate lead placement is a novel and potentially impactful approach.

Again, we would like to thank the reviewer about his feedback regarding the organization of our work.

However, several critical methodological and conceptual issues need to be addressed before this manuscript can be accepted for publication. Moreover, several technical inaccuracies, assumptions, and limitations are not sufficiently acknowledged.

We attempted to address these limitations in the order they appear in your critiques. We totally believed that this has helped to further clarify our work as well as enhancing the quality of our submission.

Major revisions:

  1. The study's sample size (n=2) is a significant limitation. The article does not sufficiently address the limitations in generalizability, especially given that only two participants were studied. SCI patients are highly heterogeneous, and conclusions drawn from such a small cohort should be tempered. The authors need to explicitly discuss how the findings from these two participants may or may not translate to the broader population of SCI patients. A deeper consideration of variability between individuals with SCI, including differences in injury chronicity, anatomical variations, and overall health status, is essential to contextualize the findings.

Thank you to the reviewer for bringing up an excellent point about the small sample size. We have revised the text to expand on the limitations in generalizability. In addition, as the reviewer suggested, we have included further context about the variability between individuals with SCI and how the results may have limits translating to the broad, diverse population. It is also important to know that most of the previous  published studies in the filed  (Harkema Group or Courtine Group) was limited to sample size less than 4 participants, because of the cost involved with implantation and training participants. We have included the following lines:

“Further, it is important to note that the SCI population is highly heterogeneous, with differences in length of time since injury, health status, and spinal cord anatomy. Due to these differences, it is important to consider that temporary mapping with MRI reconstruction may not necessarily be as effective for guiding permanent implantation of the leads for all individuals.” We believe that the translation aspect of the current work will be further enhanced by validating the current findings in a larger sample size.

  1. There is no control group for comparison, either a group receiving intraoperative mapping or a cohort using SCES without MRI guidance. While it may not be feasible to include such a group in this pilot study, the absence of a control leaves the findings relatively isolated. The authors should provide a more detailed justification for their study design choices and discuss potential confounding variables (e.g., participant age, injury chronicity, differences in rehabilitation protocols) that could affect the study’s outcomes.

We completely agree with the reviewer that a control group is vital to assess the effect of an intervention. We would like to point out that it is unfeasible to perform intraoperative mapping at our center due to the risk of infection. In addition, in our study MRI reconstruction of the spinal cord was not performed during mapping but instead retrospectively to study the effect of lead position on EMG activation of the muscles. Please see the following line:

“The initial purpose of the MRI was to confirm the location of the injury and assess the extent of injury as well as to determine the patency of the lumbosacral region before implantation. Retrospectively, MRI was then utilized as a tool to guide reconstruction of the spinal cord and ensure accurate anatomical placement of the percutaneous leads in the lumbosacral segment.”

The MRI reconstruction basically was not available at the time of implantation. We reconstructed the MRI of the spinal cord after the fact. This provided us with important information on why 0883 didn’t attain any functional goals after implantation. On top of this, it helps us to understand the effects of lead migration on functional goals in 0884.

Nonetheless, the reviewer brings up a great point and we have updated the manuscript to include further information about the study design and potential confounding variables. The following lines provide further information about the study design:

“The two participants were initially randomized into two groups: pSCES + EAW + R.T. (0884) or delayed pSCES + EAW + noRT (0883). The delayed pSCES group underwent implantation 6 months after undergoing 6 months of EAW. This latter served as the control group to determine the effects of pSCES+ EAW compared to only EAW on different measurements of the study.”

The following lines were added to the limitations section addresses the potential confounding variables that the reviewer brought up:

“The two participants’ unique attributes such as chronicity of injury, age and differences in rehabilitation protocols may have served as potential confounding variables influencing the outcomes of the study.”

  1. The interpretation of lead migration and its effect on motor function is somewhat inconsistent. The authors suggest that minor migration does not substantially affect motor function in some muscle groups, but this interpretation requires more careful analysis. The study reports "high agreement" in certain motor groups despite migration, but this is not statistically supported.

The authors must clarify whether the analysis was qualitative or quantitative, and if quantitative, more rigorous statistical methods should be applied to test for significant differences. If the comparison is qualitative, this should be clearly stated and discussed as a limitation.

We would to thank the reviewer for these excellent points. We believe that the reviewer has brought several points.

First regarding migration: in 0884 participant (Figure 6), lead migration happened in segments L3; however, the remaining electrodes remained unimpacted. This has resulted to dampen the activity of two major muscle groups (RF and HS) or altered the synergistic motor activity dynamics between these two agonist- antagonist muscle groups. However, the other segments remained unimpacted by the migration and resulted in high level of agreement in the muscle groups between mapping and re-mapping. This has led us to state this statement “ minor migration does not substantially affect motor function in some muscle groups”. However, we believe that the reviewer point correct and we have now rephrased the entire sentence.

Line 485-486: Migration clearly attenuated the motor evoked potentials of the RF and HS muscle groups and resulted in the loss of functional standing in participant # 0884. This may have potentially resulted in altering the synergistic motor activities between these two agonist-antagonist muscle groups.   However, the relative agreements were high in the other muscle groups (Table 2) between mapping and remapping that were not impacted by migration of the leads (Figure 6).

Second regarding rigorous statistical approach.  We fully agree with the reviewer that the agreement between permanent mapping and remapping need more rigorous statistical support which is limited with the sample size of n=2. To address this important point, we have added an R^2 value to measure the strength of correlation between the temporary mapping and permanent mapping motor evoked potentials. Our strategy relied primarily on identifying a visual agreement by inspection of the recruitment curves following temporary or permanent mapping. If there is a visual agreement, we then conducted an R^2 test to determine the strength of agreement. This same method was applied to test for agreement between permanent mapping and re-mapping after migration.  This analysis has been updated in the methods section in the following lines:

“Visual agreement was then further assessed by plotting data points for temporary against permanent or mapping against remapping to calculate R^2 values to examine the strength of relationship between two mapping sessions.”

The R^2 values supporting agreement were provided in the following lines in the results section:

“Visual agreement was supported by resulting R2 values for the right leg (R2: 0.81; R2: 0.85; R2: 0.88) in each case, but not for the left leg (R2: 0.02; R2: 0.07; R2: 0.21) (Table 1).”

  1. While the authors present MRI reconstruction as a reliable tool for SCES lead placement, the method is not fully validated. The current retrospective approach could be prone to errors in segmentation and alignment. The authors should provide more technical details on how the accuracy of MRI-based reconstruction was verified and discuss potential sources of error in mapping lead locations. Additionally, the claim that MRI can replace intraoperative mapping is premature without larger validation studies. The manuscript should present these findings as preliminary and call for more extensive research to confirm the efficacy of MRI-based mapping.

Thank you to the reviewer for highlighting a vital point regarding retrospective MRI reconstruction. We agree that retrospectively performing MRI reconstruction may lead to errors in segmentation and alignment. Our reconstruction of the spinal cord and determination of lead placement was performed using MRI images through MATLAB and measurements collected from DXA scans using Image J software. With regards to accuracy, we utilized available sagittal MRI slices to identify the extent of the spinal cord (with respect to the vertebrae) and axial slices were used in MATLAB to reconstruct the spinal cord after determining the thickness of each segment based on the published data [references # 35,36]. This data was coupled with lead measurements that were obtained through ImageJ to determine the lead placement with respect the reconstructed images. Our team underwent their best efforts to ensure the accuracy of reconstruction but as the reviewer brought up, limitations do exist. We have highlighted this section in detail in the limitation section.

There are multiple potential sources of error which we acknowledge are important to address as the reviewer suggested. We would like to point out that the MRI scans that were utilized for reconstruction were initially collected to identify the patency of the lumbosacral region, rather than specifically collected to perform reconstruction. In addition, spinal cord segment thichness may vary among the broad SCI population and those reported in the literature may differ, limiting the accuracy of reconstruction. The shortcomings to our reconstruction have been inserted in the limitations section of the main text in the following lines below:

“The reconstructed lumbosacral spinal cord segments may not reflect the accurate anatomical representation in both participants, because the images were not collected initially for this purpose. We primarily relied on the existing images as well as the published data to feed in our reconstruction model. Possibly, the thickness of spinal cord segments reported in the literature may vary between individuals impacting the accuracy of the developed model [35, 36]. Due to these shortcomings, it is important to note these findings should be considered as preliminary and highlight the need for further investigation into MRI-based mapping. Furthermore, the translation aspect of the current work will be enhanced by validating the current findings in a larger sample size.”

In these lines, we also note the MRI reconstruction mapping results as preliminary and call for further investigation, as requested by the reviewer.

  1. The study does not provide a clear analysis of the motor function improvements following permanent implantation. The reported outcomes focus on MEPs rather than functional outcomes like standing or stepping. While MEPs are important, the ultimate clinical goal is improved function, and the manuscript would benefit from a more detailed discussion of how the observed changes in MEPs translated (or did not translate) into meaningful improvements in motor function. The paper would also benefit from a comparison with previous studies using SCES for motor function recovery.

Thank you to the reviewer for bringing up this worthwhile point. We would like to point the reviewer to our previously published work which reports on the motor function improvements of participants implanted permanently with pSCES. There are two published papers that addressed functional outcomes of the intervention. In these studies, we have documented how the properties of the EMG muscle recruitment curves translated into participants’ ability to stand independently. This work is addressed in the following lines in the introduction:

  1. Gorgey AS, Trainer R, Sutor TW, Goldsmith JA, Alazzam A, Goetz LL, Lester D, Lavis TD. A case study of percutaneous epidural stimulation to enable motor control in two men after spinal cord injury. Nat Commun. 2023 Apr 12;14(1):2064.
  2. Alazzam AM, Ballance WB, Smith AC, Rejc E, Weber KA 2nd, Trainer R, Gorgey AS. Peak Slope Ratio of the Recruitment Curves Compared to Muscle Evoked Potentials to Optimize Standing Configurations with Percutaneous Epidural Stimulation after Spinal Cord Injury. J Clin Med. 2024 Feb 27;13(5):1344.

“Recently, we explored the potential of using the peak slope ratio to optimize the SCES configurations necessary to achieve standing in men with SCI [20] Compared to simply using the motor evoked potentials, the slope ratio of the established recruitments curves identified and narrowed down the best stimulation parameters as well as the cathodal-anodal placements in the form of wide vs. narrow-field configurations [20]. For a myriad of reasons including ASIA level and overall physical conditioning only two men out of the 4 participants were capable of achieving independent standing [20]. As a follow-up of our initial findings, we were interested to understand why two of our participants didn’t achieve functional improvement as well as why one of the two participants lost his abilitiy to stand after migration. This has led our study team to address the question of whether MRI spinal cord reconstruction could provide us with an answer to the above findings.”

The motor function improvements are also discussed in the following lines in the discussion:

“Our preliminary work demonstrated that percutaneous SCES may lead to achieving trunk control, overground standing, and restoring overground stepping in a non-functional pattern [9, 14, 23]. Two out of the four participants restored standing after successfully calculating the peak slope ratio between the extensor and flexor muscle groups necessary to maintain hip, knee and ankle extension [14].”

Additionally, in the following lines, an explanation is provided of how MRI reconstruction can explain some of the standing performance of the participants:

“However, it was unclear why the other two participants did not achieve independent standing. Similar to an earlier report [19], we successfully reconstructed the lumbosacral segments of the spinal cord using the captured axial MRI. The spinal cord reconstruction provided insights on the anatomical locations of the percutaneous leads relative to L1-S2 segments. It was clear that in 0883 participant, 62% and 75% of the contacts were not anatomically corresponding to the target segments during temporary and permanent implantation, respectively. For 0884, 63% and 75% of the contacts successfully targeted the lumbosacral segment from L3-S2. This may explain why participant 0884 managed to achieve partial standing initially after permanent implantation (despite improper coverage of L1-L2).”

Minor revisions:

  1. The statistical methods used in this study are not well described. A more detailed explanation of how the data were analyzed, especially for agreement between temporary and permanent mapping, is necessary. If visual agreement was used as the primary measure, the limitations of this approach should be acknowledged, and where applicable, quantitative methods should be employed.

Thank you to the reviewer for this important insight. We agree the statistical methods used in the study need to be expanded upon. The manuscript has been updated to include R^2 values to report on the strength of correlation between temporary and permanent mapping curves as well as between permanent mapping and migration remapping.

  1. The paper describes the use of wide-field and narrow-field electrode configurations but does not provide enough information on how these configurations were selected or optimized for each participant. More details on the rationale for configuration choices and how they were tailored to each participant’s anatomy would strengthen the methods section.

Thank you to the reviewer for bringing up a strong point about the selection and optimization of the configurations for each participant. We have added the following details about the rationale for configuration selection and optimization and how it was tailored to each participant:

“Wide-field and narrow-field configurations were selected to allow for standardization of the configurations tested to enable comparison between participants. These wide-field and narrow-field configurations were shown to successfully achieve differential modulation of spinal motor pools in previous work [Sayenko et. al, 2014]. Optimization of the configurations was highlighted in our previous work and was tailored to each participant’s unique anatomy based on an analysis of the magnitude of their individual extensor evoked potentials compared to flexors [Alazzam et. al, 2024].”

  1. Throughout the manuscript, there is some inconsistency in the use of technical terms (e.g., "motor evoked potentials," "recruitment curves," "migration"). The authors should ensure that these terms are consistently defined and used throughout the manuscript.

Thank you to the reviewer for making this crucial point. The manuscript has been updated throughout to reflect the definition of each of the terms indicated. Here are some examples of where each term is defined:

“MRI reconstruction of the lumbosacral segment also explained the differences in motor evoked potential (muscle action potential elicited by spinal cord stimulation) between temporary and permanent mapping.”

“For the purpose of the current work, the configurations tested in temporary mapping and permanent mapping were retrospectively analyzed to identify matches in anode/cathode placement, stimulation frequency, and pulse width to compare the recruitment curves (motor evoked potentials plotted versus stimulation amplitude) of different muscles between temporary and permanent mapping phase (Fig 3).”

“Despite migration (displacement of the leads from the original anatomical placement following implantation) and loss of functional standing in 0884 participant, relative agreements were high between mapping and remapping. However, migration clearly attenuated the motor evoked potentials (muscle action potentials elicited by spinal cord stimulation) of the RF and HS muscle groups.”

  1. Some of the references, particularly those related to MRI reconstruction and SCES implantation, are outdated. The authors should ensure that the most current literature is cited, particularly in the discussion of SCES technologies.

We have updated our citation for the MRI reconstruction to include the most recent literature:

This former citation was swapped out for the latter one:

Ko, HY., Park, J., Shin, Y. et al. Gross quantitative measurements of spinal cord segments in human. Spinal Cord 42, 35–40 (2004).

Fradet, L., Arnoux, P. J., Ranjeva, J. P., Petit, Y., & Callot, V. (2014). Morphometrics of the entire human spinal cord and spinal canal measured from in vivo high-resolution anatomical magnetic resonance imaging. Spine, 39(4), E262–E269. https://doi.org/10.1097/BRS.0000000000000125

Language and grammatical revisions:

  1. In the abstract, "enhance motor function to achieve standing and stepping after SCI" is awkwardly phrased. It would be more precise to say "improve motor function, aiming to restore standing and stepping abilities in individuals with SCI."

Thank you to the reviewer for this suggestion. It has been implemented into the manuscript.

  1. The phrase "Minor lead migration did not seem to impact the outcomes of spinal cord mapping" could be rephrased for clarity. Consider: "Minor lead migration appeared to have minimal impact on spinal cord mapping outcomes."

Thank you to the reviewer for this suggestion. It has been implemented into the manuscript. The entire paragraph was updated to reflect your point.

  1. In the introduction, "Research studies agreed that spinal cord mapping is an essential cornerstone step" is redundant. The phrase "essential cornerstone" should be revised for clarity, as "cornerstone" alone would suffice.

Thank you to the reviewer for this point. This edit has been made.

  1. In several sections, sentence construction is overly complex. For example, "We have chosen 8 mA because of difference in anodal locations between temporary (+14) and permanent (+15) mapping, which may have resulted in different motor evoked response" can be simplified for readability.

Thank you, the sentence was rephrased to address your concern.

  1. The discussion section occasionally lapses into passive voice, which weakens the directness of the analysis. For example, "It was unclear why the other two participants did not achieve independent standing" could be revised to "The reasons for the lack of independent standing in the other two participants were unclear."

We have corrected the sentence based on your feedback and we attempted to reduce the passive voice structures when it is possible.

Reviewer 3 Report

Comments and Suggestions for Authors

Spinal cord epidural stimulation has recently received a lot of attention as it has shown promise in inducing functional recovery in several organ systems following spinal cord injury. The major goal of the study was to determine whether MRI reconstruction of the spinal cord can be used to facilitate more accurate placement of the stimulating leads and facilitate spinal cord mapping.

Title: the term 'percutaneous' is confusing when followed by the term 'epidural'

Introduction seems a bit disorganized. It would be helpful for the reader if critical terms such as 'mapping' were explained in greater detail. For example, how is mapping performed clinically? Are the electrodes implanted in the epidural space? etc. Methods section mentions it, but the introduction section would benefit from it as well. Perhaps current paragraph 2 can follow after paragraph 3.

Figure 1 could be somewhat simplified to depict the timeline of the protocol as it pertains to this study. 

Currently from the text it is unclear why mapping cannot be done during the permanent lead placement. It would make sense for it to be dome before the trial lead placement to ensure optimal location recovery of function. Additionally,  how is the protocol of mapping, and other steps different in this study to what is routinely done in clinic. 

Finally, what are this group's recommendations regarding the future clinical practice given that the results of this study suggest that discrepancies in temporary and permanent mapping which were observed did not seem to impact the outcomes of spinal cord mapping but affected function outcomes.

Comments on the Quality of English Language

Some very minor editing is needed.

Author Response

Reviewer 3

Spinal cord epidural stimulation has recently received a lot of attention as it has shown promise in inducing functional recovery in several organ systems following spinal cord injury. The major goal of the study was to determine whether MRI reconstruction of the spinal cord can be used to facilitate more accurate placement of the stimulating leads and facilitate spinal cord mapping.

Title: the term 'percutaneous' is confusing when followed by the term 'epidural'

Thank you to the reviewer for bringing this point up. The use of the word percutaneous is used to indicate the leads are inserted through the skin into the epidural space using percutaneous needles. The route of administering the leads were percutaneous and the leads were placed in the epidural.

Introduction seems a bit disorganized. It would be helpful for the reader if critical terms such as 'mapping' were explained in greater detail. For example, how is mapping performed clinically? Are the electrodes implanted in the epidural space? etc.

Thank you to the reviewer for this clarification. We have added the following detail about mapping to the introduction to further explain it:

“Mapping refers to the process of testing various cathodal and anodal configurations to achieve targeted activation of specific spinal motor pools.”Clinically mapping is challenging process because it requires highly trained team on how to place EMG sensors on specific muscle groups to be able to develop motor evoked potentials as a result of a specific target stimulation at very low frequency of 2 Hz. This helped to develop the recruitment curves. The process itself is laborious and requires specific layers of analysis to ensure adequate data collection. Our experience that clinicians rely on visual inspection of the desired target muscle to ensure accurate stimulation. However, there is still no clinically acceptable way to determine the appropriate spinal mapping in persons with SCI.

We have also revised this sentence in the introduction to indicate the electrodes are implanted into the epidural space:

“This progression has led researchers to conclude that spinal cord stimulation (SCS), delivered by leads implanted into the epidural space, is an effective modality for restoration of motor functions after spinal cord injury (SCI) [11-15].”

Methods section mentions it, but the introduction section would benefit from it as well. Perhaps current paragraph 2 can follow after paragraph 3.

We agree with the reviewer paragraph 2 and 3 should be switched to improve the flow and have implemented this change.

Figure 1 could be somewhat simplified to depict the timeline of the protocol as it pertains to this study. 

            Thank you to the reviewer for this suggestion. We believe Figure 1 contains vital information highlighting the timeline of the protocol for 0883 and 0884 which are different and thus need to be presented in two panels. 0883 was assigned to the group with delayed implantation whereas 0884 was implanted from the beginning of the study.

Currently from the text it is unclear why mapping cannot be done during the permanent lead placement. It would make sense for it to be dome before the trial lead placement to ensure optimal location recovery of function. Additionally, how is the protocol of mapping, and other steps different in this study to what is routinely done in clinic. 

Thank you to the reviewer for this clarification. Mapping is performed after the permanent lead implantation as described in the methods section. Our institution currently does not have the capability to perform intraoperative mapping due to the risk of infection so mapping cannot be done during lead placement. As mentioned before, mapping is primarily performed for research purposes, and we are unable to report on how it may differ to treatment patients receive in the clinic (please refer to your earlier comment)

Finally, what are this group's recommendations regarding the future clinical practice given that the results of this study suggest that discrepancies in temporary and permanent mapping which were observed did not seem to impact the outcomes of spinal cord mapping but affected function outcomes.

Thank you to the reviewer for this important question. Our results simply suggest that MRI-spinal cord reconstruction is important step to ensure accurate placement of the leads. Additionally, temporary mapping, assuming accurate placement of the leads, could be simply used to guide permanent placement of the leads.

The discrepancies in mapping between temporary and permanent are highlighted in Table 1 with R2-values added to it in the revised version. The discrepancy in mapping between permanent mapping and remapping as result of migration is highlighted in Table 2 for participant 0084. The ultimate goal of mapping is to provide target segmental stimulation to achieve motor functional goal (i.e. standing).

We added the following sentence to help clarifying your point

“Migration clearly attenuated the motor evoked potentials of the RF and HS muscle groups and resulted in the loss of functional standing in participant # 0884. This may have potentially resulted in altering the synergistic motor activities between these two agonist-antagonist muscle groups.  However, the relative agreements were high in the other muscle groups that were not impacted by migration of the leads (Table 2) during the mapping and remapping sessions (Figure 6).”

Future directions should consider translation of the current work in clinical practice to determine the feasibility of this approach in a large clinical setting. We provided few future recommendations to the reader

The field of Interventional pain management has continued to evolve stronger anchors for percutaneous leads and this work should continue.  Medial migration. although less common, may also have occurred in this case before the leads had adequate time to form epidural scar.  Therefore, it is important complete an adequate waiting period is perhaps even more important for persons with spinal cord injury many of whom do not feel pain despite the presence of a healing surgical incision.

“Future strategies should consider using evolved and stronger implanted anchors to reduce migration of percutaneous leads in persons with SCI, similar to the interventional pain management trials. Medial migration, although less commonly, may have occurred before the leads had adequate time to form epidural scar. Therefore, it is important to ensure adequate recovery period, possibly 4 weeks, following permanent implantation.”

Reviewer 4 Report

Comments and Suggestions for Authors

1-      The abstract is too long and should be shortened.

2-      In my idea with 2 patients (2 sample size), authors can not claim about the satisfying of the study and validation the results. For this reason, they need to modified the hypothesis and methodology.

3-      With two samples, statistical analysis was missing in this study due to the small sample (two patients) size.

4-      The legend of Figure 1 is too long and should be shortened; some of them should be moved in the main text.

5-      In line 185, There is no unit of time for TE and TR.  “…….. (slice thickness: 3 mm, T.R.: 9350, T.E.: 102; flip angle: 150) for pre-screening purposes”.

6-      Figures 3, 7, and 8 needs to resolution improvement.

7-      There is no statistical analysis for comparing the results of two studied persons from point of agreement or disagreement view.

8-      The conclusions does not consistent with the enough evidence, which presented in the work.

Comments on the Quality of English Language

Some grammar and spelling in the English language needs correction.

Author Response

Reviewer 4

  • The abstract is too long and should be shortened.

We agree with the reviewer that the abstract could be shortened. We have cut the following lines to trim it:

“SCES mapping using a MedTronic tablet to steer lead coverage was performed on days 1-3 following temporary trial and for 2 weeks following permanent implantation.”

“However, temporary and permanent mapping agreed primarily in the wide-field configurations in both participants.”

“Discrepancies in temporary and permanent mapping may be explained by retrospective MRI reconstruction of the spinal cord which indicated that the percutaneous leads did not specifically target the entire L1-S2 segments in both participants.”

Was revised to “These may be explained by retrospective MRI reconstruction of the spinal cord which indicated that the percutaneous leads did not specifically target the entire L1-S2 segments in both participants.”

The following sentence was also deleted

“MRI reconstruction of the spinal cord determined the exact location of anticipated locomotor segments that may enhance motor evoked potentials of the stimulated muscles in persons with SCI.”

The sentence after was rephrased to:

“Temporary mapping coupled with MRI reconstruction has the potential to be considered as a guidance for permanent implantation considering target activation of the spinal cord locomotor centers.”

  • In my idea with 2 patients (2 sample size), authors can not claim about the satisfying of the study and validation the results. For this reason, they need to modified the hypothesis and methodology.

Thank you to the reviewer for bringing up this valid point about the sample size. We have added the following lines to the limitations section to contextualize the claims about the results:

“Further, it is important to note that the spinal cord injury population is highly heterogeneous, with differences in length of time since injury, health status, and spinal cord anatomy. Due to these differences, it is important to consider that temporary mapping with MRI reconstruction may not necessarily be as effective for guiding permanent implantation of the leads for all individuals. The two participants’ unique attributes such as chronicity of injury, age and differences in rehabilitation protocols may have served as potential confounding variables influencing the results.”

  • With two samples, statistical analysis was missing in this study due to the small sample (two patients) size.

Thank you to the reviewer for this valuable insight. We fully agree with the reviewer that the agreement between permanent mapping and remapping need more rigorous statistical support (which is limited with a sample size of n=2). To address this important point, we have added an R^2 value to measure the strength of correlation between the temporary mapping and permanent mapping motor evoked potentials. Our current strategy is to determine initially a visual agreement by inspection of the recruitment curves following temporary or permanent mapping. If there is a visual agreement, we then conduct an R^2 test to determine the strength of agreement to provide a more comprehensive statistical comparison. This same method was applied to test for agreement between permanent mapping and remapping after migration.  This analysis has been updated in the methods section in the following lines:

“Visual agreement was then further assessed by plotting data points for temporary against permanent and mapping against remapping to calculate R^2 values to statistically validate the strength of relationship between two mapping sessions.”

The R^2 values supporting agreement were provided in the following lines in the results section:

“Visual agreement was supported by resulting R2 values for the right leg (R2: 0.81; R2: 0.85; R2: 0.88) in each case, but not for the left leg (R2: 0.02; R2: 0.07; R2: 0.21) (Table 1).”

  • The legend of Figure 1 is too long and should be shortened; some of them should be moved in the main text.

We fully agree with the reviewer that the legend of Figure 1 could be shortened. We have deleted the following lines from the caption which are already reflected in the study design and methods section of the main text:

“After removal of the temporary leads, participants were given 3-4 weeks rest and then received the permanent implantation. Permanent mapping occurs for approximately 3 weeks. For 0803, there was a delay by design and participant was implanted after conducting 6 months of EAW; 3 days per week. 0883 participant underwent 6 months (24 weeks; phase I) of EAW without SCES.”

“After mapping, participants continued EAW+SCES (1 hour) and followed by 1 hour of standing task-specific training (StTST) in the standing frame, parallel bars or using a walker.”

“Each training day consists of one hour of spinal cord epidural stimulation (SCES) combined with EAW, followed by one hour of SCES combined with StTST.”

  • In line 185, There is no unit of time for TE and TR.  “…….. (slice thickness: 3 mm, T.R.: 9350, T.E.: 102; flip angle: 150) for pre-screening purposes”.

Thank you to the reviewer for shining light on this. The following line has been updated with the correct unit of ms.

“Before enrollment, participants underwent magnetic resonance imaging (MRI; T2 Turbo Spin Echo with long band width; SIEMENS 1.5T) with the following scanning sequence (slice thickness: 3 mm, T.R.: 9350 ms, T.E.: 102 ms; flip angle: 150) for pre-screening purposes.”

  • Figures 3, 7, and 8 needs to resolution improvement.

Thank you for pointing out the problem with low resolution. We have provided new figures with higher resolution. We have now increased the resolution of the 3 figures to 472 dpi.

  • There is no statistical analysis for comparing the results of two studied persons from point of agreement or disagreement view.

Thank you to the reviewer for bringing up this important element. We agree statistical analysis enabling comparison between the two participants would be beneficial. However, this is difficult due to the small sample size. In addition, the configurations during temporary mapping were tested at a pulse width of 500 μs in one participant (0884) and 250 μs in the other (0883), limiting comparison of the motor evoked potentials. Although, we have provided R^2 values to allow for comparison of the strength of agreement between temporary and permanent mapping for each participant.

8-      The conclusions does not consistent with the enough evidence, which presented in the work.

Thank you to the reviewer for bringing up this valid point. We have adapted the conclusion to further reflect the evidence provided in the manuscript. Please see the following updated lines in the conclusion:

“Conclusions

We attempted to identify the agreement between temporary and permanent mapping results in two individuals with SCI. During both mapping periods, various electrode configurations with different stimulation parameters were tested to achieve the optimal segmental activation of the paralyzed leg muscles. Our results showed a discrepancy in the recruitment of the leg muscles between temporary and permanent mapping which was linked to placement of the leads n as demonstrated by MRI reconstruction. Our results, therefore, suggest that temporary mapping with MRI reconstruction of the spinal cord may be used to achieve optimal activation of the spinal motor pools and guide permanent mapping during percutaneous SCES implantation. knowledge about the MRI location of the conus end is essential to determine the exact location for anatomical placement of the leads to target L1-S2 segments. Our results suggested that placing the caudal end of the leads 1.5 cm above the end of the conus end would accurately align the leads against the S2 spinal cord segment. Migration modestly dampened the activity of synergistic muscle groups that resulted in losing functional standing in a person with SCI.